# Vehicle Teleoperation: Human in the Loop Performance Comparison of Smith Predictor with Novel Successive Reference-Pose Tracking Approach

**DOI:** 10.3390/s22239119

**Published:** 2022-11-24

**Authors:** Jai Prakash, Michele Vignati, Edoardo Sabbioni, Federico Cheli

**Affiliations:** Department of Mechanical Engineering, Politecnico Di Milano, 20156 Milan, Italy

**Keywords:** latency, time delay, vehicle teleoperation, Smith predictor, NMPC, successive reference-pose tracking

## Abstract

Vehicle teleoperation has the ability to bridge the gap between completely automated driving and manual driving by remotely monitoring and operating autonomous vehicles when their automation fails. Among many challenges related to vehicle teleoperation, the considered ones in this work are variable time delay, saturation of actuators installed in vehicle, and environmental disturbance, which together limit the teleoperation performance. State-of-the-art predictive techniques estimate vehicle states to compensate for the delays, but the predictive states do not account for sudden disturbances that the vehicle observes, which makes the human-picked steer inadequate. This inadequacy of steer deteriorates the path-tracking performance of vehicle teleoperation. In the proposed successive reference-pose-tracking (SRPT) approach, instead of transmitting steering commands, the reference trajectory, in the form of successive reference poses, is transmitted to the vehicle. This paper introduces a method of generation of successive reference poses with a joystick steering wheel and compares the human-in-loop path-tracking performance of the Smith predictor and SRPT approach. Human-in-loop experiments (with 18 different drivers) are conducted using a simulation environment that consists of the integration of a real-time 14-DOF Simulink vehicle model and Unity game engine in the presence of bidirectional variable delays. Scenarios for performance comparison are low adhesion ground, strong lateral wind, tight corners, and sudden obstacle avoidance. Result shows significant improvement in reference tracking and in reducing human effort in all scenarios using the SRPT approach.

## 1. Introduction

Vehicle teleoperation is an activity of driving a vehicle by sending driving commands to the vehicle from a control station (that is stationary) and, in most cases it is far away from the vehicle. In the case of a wireless communication channel, potential civil applications of vehicle teleoperation can be last-mile delivery of rental/shared vehicles, human remote assistance in the event of autonomous vehicle failure, valet parking, etc. It can also be useful in military application, including rescue, reconnaissance and patrol missions. The 4G LTE wireless broadband connection is the best candidate for the data communication protocol in vehicle teleoperation due to its widespread availability around the world.

The advent of autonomous vehicles is likely to transform individual and public transportation [1] as well as freight transport and intralogistics [2,3]. Autonomous vehicles may perform well in the scenarios above, but they may still struggle in critical traffic circumstances that a human could easily handle, such as parking lots, pedestrian crossing areas, or construction roads. Vehicle teleoperation may be able to assist in the transition from human-driven to autonomous vehicles. Most military robotic systems, according to Cosenzo and Barnes [4], will require active human control or, at the very least, supervision, with the capacity to take over if necessary. This application also comes under the use cases of vehicle teleoperation. Vehicle teleoperation can also be beneficial to human driver safety because the human operator is in the control station and not inside the vehicle.

Vehicle teleoperation also has shortcomings; the human operator is often constrained to operating the vehicle using 2D video feeds with limited resolution and field of view (FOV) [5]. Additionally, time delay in data processing and transmission across a wireless network causes the video stream and driving command to be hundreds of milliseconds late. Time delays reduce the speed and accuracy with which human operators can accomplish a teleoperation task [6,7]. When delays are considerable, human operators have a tendency to overcorrect steer, causing oscillations that might degrade teleoperation performance and possibly destabilize the control loop [8,9].

### 1.1. Related Work

Predictive display and waypoint-based vehicle teleoperation are the two major categories in the literature. In human-in-the-loop experiments, predictive displays have been shown to be effective in compensating delays and improving vehicle maneuverability [10,11,12,13,14,15,16,17]. To estimate the vehicle position, it takes into account the delay in the control loop as well as the driving commands issued by the human operator. One way to convey the predicted position of the vehicle to a human operator is by displaying a “third-person view” of the expected position [14]. The prediction model can either be model-based [13], model-free [15], or a blend of both [16]. A vehicle model is necessary to forecast the vehicle response in model-based predictors, and the prediction accuracy is dependent on the vehicle model’s correctness. Prediction accuracy deteriorates in the presence of unforeseen disturbances in the driving environment, such as a low-adhesion road or a cross wind. In the model-free approach [15], delayed state dynamics received from the vehicle are used to make the prediction. It suffers from an another delay due to the convergence time involved in the state prediction. The blending of both approaches results slightly improved operation. Alonzo Kelly et al. [17] estimated the vehicle predicted position based on a velocity-driven non-linear vector differential equation assuming con. Even though it makes no attempt to simulate the forces of ground contact and traction, it has shown to be more than adequate for replicating the network delays of less than a second. Smit Saparia et al. [18] introduced an active safety system for collision avoidance in vehicle teleoperation. It receives the usual steering and reference velocity commands from the control station, but modulates them as per the potential fields of the obstacles. Sterling J. Anderson et al. [19] also retained steer-based driving in vehicle teleoperation. Their approach relies on path homotopies identification, which satisfy a set of position constraints, bounding a heuristically optimal or driver-preferred set of collision-free paths. The control authority of vehicle steer is allocated between the driver and obstacle collision threat minimization by a means weighted average. Vehicle steer is transmitted to the driver using visual or steering torque feedback to inform him/her about the predicted and action already being taken by the vehicle. However, this approach attaches an additional safety layer, and the robustness of teleoperation under the presence of delay is not discussed. In brief, predictive displays attempt to counter the time delay in loops by anticipating states using delayed states as input. This is useful for human-in-loop teleoperation, as it allows the human operator to not have to wait for the feedback and provides the sense of controlling the vehicle in real time. The drawback is that when prediction accuracy declines, the chances of asynchrony rise.

In waypoints-based driving (shared/cooperative control), vehicle control is based on automated driving along predefined paths. This eliminates network time delay out of the control loop. Michael Fennel [20] proposed an offline path follower, where the operator is not actively controlling the maneuver but just supervising. Additionally, the generation of a path is an extra task to be performed by the control station. Macharet and Florencio [21] proposed a navigation system that employs artificial potential fields and vector field histograms, with a human operator in the loop. Sen Zhu et al. [22] introduced a touchscreen-guidance-point-based cooperative control for vehicle teleoperation. The set of guidance points (in navigable regions) are generated by the vehicle-perception system. Subsequently, the vehicle autonomous control system generates the desired trajectory based on the kinematic constraints of the vehicle. As the steering commands is being generated by the vehicle control system, the network delay is out of the control loop, which stabilizes the loop. However, compared to natural steer-based driving, this approach involves touchscreen-based path selection.

The shared-control approach in vehicle teleoperation still possesses some potential improvements. Many prior works considered navigation only in structured environments. In circumstances such as an autonomous navigation based on a road network, global path points can be selected in advance. In unstructured environments, such as off-road places, often, it is not reasonable to assume that the navigation control system knows the prominent homotopies to reach the goal position.

### 1.2. Contribution of Paper

It is important to note how the current publication differs from the past and ongoing research given the extensive body of work that precedes the work provided here. Despite using sophisticated position-predicting techniques, predictors suffer from prediction errors and environmental disturbances (such as low-adhesion surface and cross wind) that the real vehicle observes, as the after-effect of these disturbance are seen by the human operator after a delay.

In our previous work [23], we introduced a position-based control strategy, SRPT, for vehicle teleoperation to account for environmental disturbances. In this strategy, the upcoming reference-pose to be followed by the vehicle is transmitted (∼30 Hz) discretely to the vehicle, which makes the vehicle aware of the reference pose to be reached. Consequently, the vehicle controller takes into account the actuator constraints and environmental disturbance (if any), to optimize for the steer and speed command. The reference-pose decider in the control station is a mathematical model of a driver, which accounts for the already known mission plan and the received delayed vehicle position.

In the proposed SRPT approach, the control station transmits reference poses instead of steer and speed commands. To accommodate for humans in the operation loop, first, this paper proposes a steering-joystick-based method to generate successive reference poses for the SRPT approach. The successive poses correspond to the target position to be reached by the vehicle in the immediate future. Later, to assess the performance improvement, human-in-loop vehicle teleoperation experiments are conducted using a simulation environment (Figure 1). It consists of integration of a real-time 14-DOF Simulink vehicle model and Unity game engine through UDP (User Datagram Protocol) data transmission, incorporating bidirectional variable delays. Two scenarios are considered: the first consists of non-traffic but progressively increasing difficult maneuvers, and the second consists of getting around sudden obstacles. The importance of this work is it that it keeps the usual steer-based physical interface for the human operator, while converting the steer reference to pose reference for the real vehicle. This, in addition to compensating for environmental disturbances, also allows the human operator to compensate for the under/over-steer nature of the vehicle, eventually reducing cognitive load as per the literature [24]. The next importance is performance comparison with the Smith predictor using human-in-loop drive experiments in the presence of disturbances/sudden obstacles, which exhibits the performance benefits of the SRPT approach.

### 1.3. Outline of Paper

The rest of the paper is organized as follows. Section 2.1 presents the characteristics of network delay. Section 2.2 explains the modes of vehicle teleoperation simulated for comparison and the explanation of the novel SRPT mode. Section 3 provides an overview of the simulation platform. Section 4 discusses the experimental structure. Section 5 presents and discusses the results. Section 6 concludes with the work summary, key findings, and aspects to be considered for virtual to real-world replication.

## 2. Method

This section first discusses the characteristics of the network delay observed. Then, it explains the various modes of vehicle teleoperation considered.

### 2.1. Uplink Delay and Variable Downlink Delay Characteristics

From the control station perspective, the time delay involved in vehicle teleoperation can be divided into two parts. One is downlink delay ($\tau_{2}$), which is associated with the streamed images received by the control station. The other is uplink-delay ($\tau_{1}$), which is associated with the delay between the generation of driving commands at the control station and the actuation of those at the vehicle. $\tau_{2}$ can be considered a lumped sum of camera exposure delay, image encoding time consumption, network delay in transmitting the images toward the control station and image-decoding time consumption. $\tau_{1}$ can be considered a lumped sum of network delay in transmitting the driving commands towards the vehicle, and vehicle actuation delay. In the case of wireless communication using 4G, variability is associated with both downlink and uplink delays. Figure 2 presents both sided delays with corresponding utilized bandwidth. The information corresponds to 5000 picture frames and driving commands. This test is carried out in a typical urban setting, with the vehicle connected to 4G mobile connectivity and the control station via wired LAN to the internet. $\tau_{1}$ is measured at the vehicle by subtracting the timestamp of driving commands from the current timestamp, and $\tau_{2}$ is measured at the control station by subtracting the timestamp of an image received from the current timestamp.

Uplink delay is unknown and downlink delay is known at the control station. As the amplitude and variability of uplink delays are smaller, a constant high stochastic value (an approach adopted in [13]) of 60 ms is used in the following teleoperation simulations. For downlink delay, generalized extreme value distribution, GEV(ξ=0.29,μ=200,σ=9) is used [13,16]. Here, ξ is the shape parameter, μ is the location parameter and σ>0 is the scale parameter. Positive ξ means that the distribution has a lower bound $(\mu -\frac{\sigma }{\xi })\approx 169$ ms (>0) and a continuous right tail based on the extreme value theory. During a stable 4G connection (Figure 2), data packets received are found to be in FIFO order. This indicates no data loss and FIFO queue behavior of the communication.

### 2.2. Vehicle Teleoperation Modes

To evaluate the benefits of the proposed method in compensating for network delay and disturbances, four driving modes are considered and compared in the simulation testing environment:

#### 2.2.1. No-Delay Mode

This mode is closest to driving a vehicle from the driver seat. As delays are absent in this mode, the human operator perceives the vehicle undelayed state and his/her control actions are applied to the vehicle instantly.

For this mode, delays ($\tau_{1};\tau_{2}$) are zero. Relating mode schematic (Figure 3) with simulation infrastructure (Figure 1), the Simulink 14-dof vehicle model receives the steer from the steering joystick and outputs the vehicle position to the Unity visual interface through UDP packets. A PI cruise control placed just before the 14-dof vehicle model maintains the vehicle speed near the reference speed (22 km/h).

To make the human quickly aware of the steering ratio, a steer indicator (Figure 4) based on kinematic vehicle motion is overlayed on the visual. The indicator is in the form of a rectangular outline of the vehicle. Its 2D-Pose ($x_{\mathrm{ind}\;},y_{\mathrm{ind}\;},\psi_{\mathrm{ind}\;}$) relative to the base pose (Figure 4) is given as
(1)$$\begin{array}{cc} \psi_{\mathrm{ind}\;} & =\frac{L_{\mathrm{ind}\;}}{R}\\ x_{\mathrm{ind}\;} & =R\sin\psi_{\mathrm{ind}\;}-l_{F}\left(1-\cos\psi_{\mathrm{ind}\;}\right)\\ y_{\mathrm{ind}\;} & =R\left(1-\cos\psi_{\mathrm{ind}\;}\right)+l_{F}\sin\psi_{\mathrm{ind}\;} \end{array}$$
where *R* is the Ackermann radius given by
(2)$$\begin{array}{cc} R & =\frac{L}{\tan\delta }\;\;\;\;\;\;\;\;\;\;\left\{\begin{array}{cc} \delta =1e^{-3} & ;\;\mathrm{if}\;\delta ==0\\ \delta & ;\;\mathrm{otherwise}\; \end{array}\right\} \end{array},$$
where δ is the front axle steer; $L_{\mathrm{ind}}$ is the ahead-distance of the indicator; $l_{F}$ is front axle distance from the CG; *L* is the wheel base. As the purpose of the steer indicator is just to indicate the neutral steering motion, a short enough distance ($L_{\mathrm{ind}}=4$ m) which makes the indicator visible after the vehicle bonnet is considered. The distance is constant for no-delay, delay and Smith modes.

As delays are absent in the no-delay mode, the steer-indicator 2D pose is relative to the undelayed vehicle pose received by Unity (Figure 4 and Figure 5a).

#### 2.2.2. Delay Mode

This mode represents typical delayed vehicle teleoperation, where $\tau_{1}$ and $\tau_{2}$ in Figure 3 are given in Section 2.1. To simulate this, the Simulink 14-dof vehicle model receives delayed steer, and Unity receives delayed vehicle pose. As a result, the human operator observes delay in his/her control actions and respective visual feedback. The steer indicator (inside Unity) still receives undelayed steer from human, but as the vehicle pose received (from the Simulink vehicle model) is delayed, it is only able to overlay the indicator respective to the delayed vehicle pose (Figure 4 and Figure 5b).

#### 2.2.3. Smith-Predictor Mode

Schematic of the Smith predictor for the vehicle teleoperation control loop is shown in Figure 6 for known time delays.

The steer input (δ) is passed through a local predictor model ($P^{\prime }$) of the vehicle, which then passes through $\left(1-e^{-\left(\tau_{1}+\tau_{2}\right)s}\right)$, where a time-delayed version of the output is subtracted from the real-time version. With this schematic, feedback ($X_{p}$) given to the human operator is
(3)$$X_{p}=P^{\prime }\delta \left(1-e^{-\left(\tau_{1}+\tau_{2}\right)s}\right)+\delta e^{-\tau_{1}s}Pe^{-\tau_{2}s}$$
which, in turn, if the predictor model ($P^{\prime }$) is equal to vehicle model (*P*) becomes
(4)$$X_{p}=P\delta$$

It provides the human operator the sense of controlling the vehicle in real time. If $P^{\prime }=P$, the transfer function of the closed-loop delayed system is
(5)$$\frac{X}{X_{REF}}=\frac{HP}{1+HP}e^{-\tau_{1}s}$$

In reality, *P* and $P^{\prime }$ are different. In our simulation environment, *P* is a non-linear 14-dof vehicle model, while $P^{\prime }$ is non-linear single-track vehicle model provided by Simulink block Vehicle Body 3DOF Single Track [25].Block inputs are velocity and steer, while its constant parameters are single-track vehicle dimensions and cornering stiffnesses (Table 1).

The Smith predictor bypasses the delay in the observation and transforms the system into a pure forward-delay system (Equation (Equation 5)). The system still tracks the input with a constant forward delay offset; however, this will not affect the controllability of the vehicle. Further explanation is presented in previous work [23].

A problem with the previous delay-mode is that the steer indicator is respective to the delayed vehicle pose. To rectify it, Smith mode predicts the undelayed vehicle pose ($X_{P}$). Consequently, the steer indicator is overlayed relative to the predicted vehicle pose (Figure 5c) instead of to the delayed vehicle pose.

#### 2.2.4. SRPT Mode Using NMPC

In the above approaches, the control station transmits steer commands to the vehicle. In this SRPT approach, reference poses are transmitted to the vehicle. To generate reference poses, the idea of the steer indicator (previously discussed in Figure 4 and Figure 5) is extended and used. Instead of having a constant ahead distance ($L_{ind}$) for the indicator, the below relation is used for it:(6)$$L_{ind}=V\cdot \tau +\max\left(V\cdot \Delta t_{Horizon},\;\;l_{F}\right)$$

Here, *V* is the vehicle speed, $\tau =\tau_{1}+\tau_{2}$ is the round trip delay, and $\Delta t_{Horizon}=1s$ is the NMPC horizon that acts as the look-ahead time for the steer indicator. The first term in Equation (Equation 6) accounts the delay in the control loop. The second term in Equation (Equation 6) refers to the terminal of the NMPC horizon. The max condition ensures non zero $L_{ind}$ at zero speed. The use of a steer indicator is multifaceted in this approach. The first use is for the indication of steer, and the second is to generate reference poses. Referring to the schematic of the SRPT approach (Figure 7), Unity (control-station) overlays the steer indicator relative to the delayed vehicle pose (Figure 4 and Figure 5d). As the control station also receives delayed vehicle pose, $X\left(t\right)e^{-\tau_{2}s}$ in global reference frame, the global pose of the steer indicator can be calculated. Eventually the global pose of the steer indicator, $X_{Ref}\left(t+\tau_{1}+\Delta t_{Horizon}\right)$, is transmitted to the vehicle, which act as a reference pose for the vehicle.

NMPC Prediction model: On the vehicle side, the NMPC block receives the reference poses, and by analyzing the vehicle current states, actuator constraints and environmental disturbances, it optimizes for steer-rate and acceleration commands [23].

A single-track model (Figure 8) with states
(7)$$x={\left[\;\beta ;\;\dot{\psi };\;\psi ;\;F_{y,F};\;F_{y,R};\;x;\;y;\;\delta ;\;V_{x}\right]}^{T},$$
and dynamics given below is used in the NMPC prediction model:(8)$$\dot{x}=\left[\begin{array}{c} \frac{1}{mV}\left(F_{y,F}\cos\delta +F_{x,F}\sin\delta +F_{y,R}\right)-\frac{\beta \cdot a}{V}-\dot{\psi }\\ \frac{1}{I_{Z}}\left[\left(F_{y,F}\cos\delta +F_{x,F}\sin\delta \right)l_{F}-F_{y,R}\;l_{R}\right]\\ \dot{\psi }\\ \frac{V}{\lambda }\left[F_{y,F,ss}-F_{y,F}\right]\\ \frac{V}{\lambda }\left[F_{y,R,ss}-F_{y,R}\right]\\ V\cos(\psi +\beta )\\ V\sin(\psi +\beta )\\ \dot{\delta }\\ a \end{array}\right].$$

In this work, states (x) are considered known, as they are a simulation of a 14-dof vehicle model. However, in the physical experiment, a state estimator has to be present for state estimation, where the measurement of [$\dot{\psi }$; lateral acceleration] from IMU and [$\delta ;V_{x}$] from encoders can be accounted for in the correction step.

Accounted longitudinal forces (Equation (Equation 9)) consist of inertial force, rolling force, and aerodynamic force. During acceleration, a major part of the traction is provided by the front axle due to the front-wheel-drive (FWD) vehicle. During deceleration, the braking force is divided among the axles as per the braking bias distribution. For simplicity, longitudinal dynamics without relaxation length phenomenon is considered and thus given by
(9)$$\begin{array}{cc} \; & F_{x,F}=\left\{\begin{array}{cc} \;\;\;\;\;\;\;\;\;\;\;\;m\;a+f_{V}\;m_{R}\;g+C_{Aero}\;V^{2}, & \mathrm{if}\;a\geq 0\\ \;\;\;\;\;\;\;\;\;\gamma \;(m\;a+f_{V}\;m\;g+C_{Aero}\;V^{2}), & \mathrm{otherwise} \end{array}\right.\\ \; & F_{x,R}=\left\{\begin{array}{cc} \;\;\;\;\;\;\;\;\;\;\;\;\;\;\;\;\;\;\;\;\;\;\;\;-f_{V}\;m_{R}\;g, & \mathrm{if}\;a\geq 0\\ \left(1-\gamma \right)\;\left(m\;a+f_{V}\;m\;g+C_{Aero}\;V^{2}\right) & \mathrm{otherwise}. \end{array}\right. \end{array}$$

Considering the non-linear saturation model for tire force characteristics, [26]
(10)$$F_{i,ss}=D_{i}\;\mathrm{atanh}\left(C_{i}\;B_{i}\;\sigma \right),$$
knowing longitudinal forces from Equation (Equation 9), independent longitudinal slips are given by
(11)$$\begin{array}{c} \sigma_{x,F}≃\frac{\mathrm{atanh}\frac{F_{x,F}}{D_{x,F}}}{B_{x,F}C_{x,F}}\;\;\;;\;\;\;\sigma_{x,R}≃\frac{\mathrm{atanh}\frac{F_{x,R}}{D_{x,R}}}{B_{x,R}C_{x,R}} \end{array},$$
and independent lateral slips are given by
(12)$$\begin{array}{c} \sigma_{y,F}≃\tan\delta -\beta -\dot{\psi }\frac{l_{F}}{V}\\ \sigma_{y,R}≃\;\;\;\;\;\;\;\;\;\;\;-\beta +\dot{\psi }\frac{l_{R}}{V} \end{array}.$$

Net slips are given by
(13)$$\begin{array}{c} \sigma_{F}=\sqrt{\sigma_{x,F}^{2}+\sigma_{y,F}^{2}}\\ \sigma_{R}=\sqrt{\sigma_{x,R}^{2}+\sigma_{y,R}^{2}} \end{array}.$$

Steady-state lateral forces used in Equation (Equation 8) are given by
(14)$$\begin{array}{c} F_{y,F,ss}=\frac{\sigma_{y,F}}{\sigma_{F}}\;D_{y,F}\;\mathrm{atanh}\left(B_{y,F}\;C_{y,F}\;\sigma_{F}\right)\\ F_{y,R,ss}=\frac{\sigma_{y,R}}{\sigma_{R}}\;D_{y,R}\;\mathrm{atanh}\left(B_{y,R}\;C_{y,R}\;\sigma_{R}\right) \end{array}.$$

The outputs of the NMPC optimization routine are, *a*, the vehicle acceleration, and $\dot{\delta }$ the steer angular velocity. To make it applicable for zero vehicle speed, wherever the *V* is in the denominator in Equation (Equation 8), it is substituted by max(0.01,V).

Parameters used in the NMPC block correspond to a FWD typical passenger vehicle and mentioned in Table 1.

NMPC objective function: The aim is to keep the vehicle motion along with the received successive reference poses. The clothoid path is a preferred path in the vehicle motion because it resembles natural driving where steer changes linearly. Eliou and Kaliabetsos [27] suggest that cubic spline can be a first approximation for the clothoid curve. Each received reference pose (Figure 7), $X_{Ref}\left(t+\Delta t_{Horizon}\right)$ which is in the global reference frame, is transformed into the vehicle reference frame $\left[x_{Ref};\;y_{Ref};\;\psi_{Ref}\right]$, on which a local cubic spline is estimated (Equation (Equation 15), Figure 9, [23]). The spline maintains G1 continuity at both ends, i.e., starting tail with β and end tail with $\psi_{Ref}$. Note that spline coefficients update before every prediction horizon:(15)$$y=Ax^{3}+Bx^{2}+Cx+D$$

The terminal cost of NMPC is such that the predicted vehicle pose at the end of the horizon lies closer to the spline. The stage cost contains NMPC outputs $\left[\dot{\delta };\;a\right]$ and deviation from the reference vehicle speed. The net cost formulation over prediction horizon, *N*, is given by
(16)$$\begin{array}{cc} \min\; & \sum_{i=0}^{N-1}U_{i}^{t}\;R\;U_{i}+\sum_{i=0}^{N-1}X_{i}^{t}\;Q\;X_{i}+X_{N}^{t}\;P\;X_{N} \end{array}$$
where,
(17)$$\begin{array}{ccc} U_{i} & =\left[\begin{array}{c} \dot{\delta }\\ a \end{array}\right]\;\forall i\in \left[0,N-1\right] & \end{array}$$
(18)$$\begin{array}{ccc} X_{i} & =\left\{\begin{array}{c} \left[V_{Ref}-V_{i}\right]\;\forall i\in \left[0,N-1\right]\\ \left[\begin{array}{c} Ax_{i}^{3}+Bx_{i}^{2}+Cx_{i}+D\;\;\;\;\;\;-\;y_{i}\\ \tan^{-1}\left(3Ax_{i}^{2}+2Bx_{i}+C\right)-\psi_{i} \end{array}\right]\;\forall i=N \end{array}\right. & \end{array}$$


Here, $\left[x_{i};\;y_{i};\;\psi_{i}\right]$ are vehicle poses in prediction horizon (in vehicle reference frame shown in Figure 9). NMPC cost penalties are summarized in the following table (Table 2).

NMPC constraints: Steer-rate constraint (actuator limitation) and acceleration constraint (for passenger comfort) are
(19)$$\begin{array}{c} \left[{\dot{\delta }}_{min}=-20^{\circ }/s\right]\leq \dot{\delta }\leq \left[{\dot{\delta }}_{max}=+20^{\circ }/s\right]\\ \;\;\;\;\left[a_{min}=-3\;m/s^{2}\right]\leq a\leq \left[a_{max}=1\;m/s^{2}\right] \end{array}$$

Output constraints (Equation (20)) are the maximum steer, non-negative vehicle velocities, and tire friction utilization constraint, which prohibit the vehicle from utilizing road friction beyond a threshold ($\mu_{cons}$). This eventually limits longitudinal and lateral accelerations. Considering a typical urban environment, a conservative friction limit, $\mu_{cons}=0.3$ is assigned to maintain the tire characteristics in the linear region for the majority of road conditions:(20)$$\begin{array}{c} \left[\delta_{min}=-25^{\circ }\right]\leq \delta \leq \left[\delta_{max}=+25^{\circ }\right]\\ 0\leq V\\ \frac{{∥\left(F_{x,F},\;\;F_{y,F,ss}\right)∥}_{2}}{m_{F}\;g}\leq \mu_{cons}\\ \frac{{∥\left(F_{x,R},\;\;F_{y,R,ss}\right)∥}_{2}}{m_{R}\;g}\leq \mu_{cons}. \end{array}$$

The optimal control problem is discretized in N=50 intervals using discrete multiple shooting and solved by sequential quadratic programming using the real-time NMPC solver ACADOS [28,29].

## 3. Simulation Platform

A real-time driver-in-the-loop simulation test platform is developed using MATLAB Simulink + Unity3D to emulate the network delayed vehicle teleoperation system, as shown in Figure 1. The human operator generates steer commands using the Logitech G920 steering wheel, which is connected to Simulink and as well as to Unity. The Simulink output is connected to Unity to provide visual feedback through monitors. The data flow, respective to different teleoperation modes, is mentioned in Section 2.2.

A snapshot of the visual driving interface is shown in Figure 10. It displays the first-person view that is captured from a camera placed on the vehicle roof in the Unity environment and some other useful information too. The camera has a horizontal and vertical field of view (FOV) of $120^{\circ }$ and $80^{\circ }$ respectively. The main vehicle model consists of a 14-dof vehicle model and Pacejka tire model with parameters of a passenger vehicle. The powertrain is customized to be treated like an electric motor vehicle. The simulation runs using ODE4 solver with a fixed time step of 1 ms. The Simulink ‘simulation pace block’with sync sample time of 10 ms is included to slow down the simulation to real time. This simulation platform is not set for distributed simulation. Instead, both the driver side and the vehicle side are simulated in the same Simulink model on one computer, and signals are transmitted internally without a physical communication network. Therefore, simulated delays, either constant or varying, are added optionally to simulate a teleoperated vehicle system with the delays of interest as discussed in Section 2.1.

To simulate downlink delay, delayed vehicle pose is passed to Unity (Figure 7 and Figure 10). Unity places the vehicle on delayed pose which simulates delayed visual feedback to the human.

## 4. Human-in-the-Loop Experiments

Human-in-the-loop experiments were performed using the aforementioned simulation platform to compare the performance of vehicle teleoperation with various modes mentioned in Section 2.2 under varying delays. The experiment design, including the task and scenarios, test setup, test procedures, and analysis methods, are described in detail in this section.

### 4.1. Experiment Design

#### 4.1.1. Scenario 1

A test track of 438 m length is created in the Unity3D environment, as shown in Figure 11. It has six interesting regions, namely A–F.

A–B is cornering, C is double lane change, D is cornering on low-adhesion (wet) road, E is strong lateral wind with Chinese hat profile [30,31] and F is slalom. The task is to follow the track center line. To reduce the human learning effect, during training sessions, volunteers are asked to drive in a reversed version of the track (F to A).

#### 4.1.2. Scenario 2

A straight track of 16m length, as shown in Figure 12, is used to simulate the avoidance of sudden obstacles, such as falling cargo from a truck. The task is to circumvent the obstacle and return to the previous lane (purple centerline) without entering into the opposite lane. As this is sudden obstacle performance testing, no training session is considered. Volunteers are asked to perform it after scenario 1.

The reference vehicle speed for all the scenarios is same, $V_{Ref}=22$ km/h. It is chosen as such to have instances where required the steer rate approaches its limit of ${\dot{\delta }}_{lim}=\pm 20^{\circ }$/s. Thus, each experiment contains 4 (modes) × 3 (laps for each mode) runs over scenario 1 + 4 (modes) runs over scenario 2. Each volunteer must complete a total of 16 (12 + 4) test runs. To lessen the learning effects on a particular scenario or one mode, the order of the runs is randomly assigned in an even distribution (Figure 13). In particular, no mode is tested more than once in a row.

### 4.2. Test Procedure

Before the test, volunteers filled out an informed consent form and provide some basic information about themselves, including their age, driving years, kilometers driven in last 1 year, and their level of experience driving in a virtual environment. The runs for each volunteer are divided into training (∼15 min) and testing (∼30 min) sessions. Training session is intended to make volunteers familiar with the track, vehicle response, teleoperation modes, and delays present in the teleoperation loop. Since driving behavior varies person to person, to attain at least one similarity among the volunteers, they were verbally instructed to try to keep the steer indicator (Figure 5) aligned with the track center line while driving. They were not informed of the whole performance metrics, which in addition to consisting deviation from the center line, also consists of steer effort and task completion time. For scenario 1,training was conducted with each driving mode until the metrics between runs showed consistent performance and the volunteer felt comfortable with the mode.

## 5. Results and Discussion

The simulation was run on a Desktop computer with Intel i7-7800X CPU core frequency of 3.5 GHz supplemented with Nvidia GTX 1050Ti GPU. Among 18 volunteers (all researchers), 2 volunteers had no car driving experience. One inexperienced driver (racing video game player) completed the runs successfully while the other (not a racing video game player) struggled in following the path given the difficult maneuvers. Therefore, the data of only 17 volunteers are analyzed below. Some of the related information of the volunteers are the following:Age group 26–31 years;Driving experience of 7–11 years (except the 1 inexperienced driver);Seven of them are familiar with racing video games.

### 5.1. Scenario 1

It consists of 6 different regions namely A–F. Here, the performance indices to be monitored are rms and max deviation from the track center-line, rms and max steer, rms steer-rate, and completion time.

Since the difficulty level of each region is different, the performance indices are calculated region-wise. Root mean square (rms) for a particular region is computed over the traveled distance along the center-line, as given below
(21)$$\begin{array}{c} Y_{\mathrm{rms}}=\sqrt{\frac{1}{D_{i,end}-D_{i,start}}\int_{D_{i,start}}^{D_{i,end}}{\left[Y\left(D\right)\right]}^{2}\;dD} \end{array}$$
where

*Y*—is the performance index to be observed.

*D*—is the distance traveled along the reference trajectory.

[i,start],[i,end]—indicates the start and end of the particular region, *i*.

Due to a prior training session, no improvement was found in consecutive laps for the same teleoperation mode. Thus in the following data analysis, all three laps for each mode, are considered to carry equal weightage. Figure 14 presents the region-wise observed lateral deviation vs. steer by the human in the experiment. In ‘Region-A’ sub-figure, the data points correspond to the rms lateral deviation and rms steer angle realized in region-A for all laps and by all drivers. Data points closer to the origin represent improved performance, as they signify lower lateral deviation from the track center line and lower steer requirement.

Red points which represent ‘Delay + no control’ mode show poor performance. This is anticipated in a teleoperated driving activity with delays as overactuation in the form of oversteering and repeated corrections [32]. ‘NoDelay’ mode shows better performance, which is expected, as the delays are absent. ‘Delay + Smith control’ lying between the ‘NoDelay’ and ‘Delay + no control’ mode, shows significant performance improvement. The ‘Delay + SRPT’ mode shows the best performance, compared to all modes. A similar trend and performance improvement with the ‘Delay + SRPT’ mode are observed in all of the regions of the track. Lateral deviation is always significantly reduced, with a decrement in steer effort too, especially in regions D-E-F, which are relatively aggressive maneuvers.

Figure 15 presents the mean and standard deviation of lateral-deviation data points presented in Figure 14. SRPT mode always shows the lowest standard deviation, which means high repeatability.

Figure 16 presents a holistic overview of the enhancement of performance indices among the teleoperation modes in all regions of the track. The completion time index is represented as ‘T’. Desired characteristics are lesser lateral deviation, lesser steer effort and lesser completion time, which means that the smaller the shaded region in the spider plot, the better the performance. The blue shaded region of SRPT mode encompasses smaller area for all the regions, particularly for regions D-E. The attribute of the SRPT mode over other modes is the regulation of vehicle speed. The SRPT mode resulted in a slight increase in the completion time, which is because, for better reference path tracking, it moderates the vehicle speed when required. For scenario 1, apart from the slalom (region F), the increment in completion time is insignificant in other regions (Figure 17). Considering the intricacies of the slalom region, an increase in completion time is justified for improved path-tracking performance (improved safety).

Table 3 reports the performance improvement of the two prime performance indices, lateral deviation and steer effort, considering the ‘Delay + no Control’ mode as reference. The performance improvement for a given mode is calculated as a percentage reduction in index compared to the reference mode as given below:(22)$$Performance\;improvement=\frac{r_{\mathrm{Ref}\;}-r_{\mathrm{mode}\;}}{r_{\mathrm{Ref}\;}}\times 100\%.$$
where *r* is the mean of the respective index for the particular region of the track. For both performance indices, the SRPT mode reports better performance improvement.

### 5.2. Scenario 2

Here, the performance indices to be monitored are the minimum safety distance from the sudden obstacle and rms of lateral movement from the track center line. The safety distance relates to the distance margin with which the vehicle is able to evade the collision with obstacle. The rms of lateral movement relates to the controllability of the vehicle. More lateral movement means more chances to enter the opposite lane, and higher difficulty to return to the straight reference path.

Figure 18 shows the experiment result of 1 of the 17 drivers. The SRPT mode trajectory is nearer to the no-delay mode compared to the other modes. This means better control over the teleoperation. This is possible because of the automatic speed moderation (by NMPC block) in the SRPT mode when rapid steering is observed.

Figure 19 shows the mean safe distance recorded for all 17 drivers. The SRPT mode results in a safe distance that is comparable and even slightly more than that in the no-delay teleoperation mode. It means that the vehicle passes farther from the obstacle, thereby being safer.

Figure 20 shows 17 data points for each mode, which belong to 17 full runs performed by all driver for each mode. Again, lateral movement corresponds to the distance from the straight center line, and completion time corresponds to the time taken to traverse the 150 m path with 3 obstacles. The no-delay data points show the best performance, as they carry less lateral movement and less completion time. Keeping safety the priority, the SRPT mode outputs comparable lateral movement at the cost of increment in completion time, while the rest of the modes show increments in the lateral movement, indicating risk in entering in the opposite lane and less controllability.

## 6. Conclusions

This article presents human-in-the-loop evaluation of the SRPT approach (previously conceptualized in [23]) in vehicle teleoperation. It assesses its ability to improve vehicle drivability/controllability in a reference path following task and safety assessment in a sudden-obstacle-avoidance scenario. The SRPT approach relies on transmitting successive reference poses to the vehicle instead of transmitting steer commands in classical approaches. The problem of generating reference poses in live vehicle teleoperation is attempted to be solved by an innovative method of kinematic steer indicator. The successive reference-poses are later received by the vehicle, where the NMPC block optimizes for steer and acceleration commands. These reference poses represent the driver intent for the maneuver. The NMPC block uses the ACADOS framework, which shows the capability to run at 50 Hz, as the mean computation time observed is 8 ms. The optimization routine penalizes the cross-track error and eventually decelerates the vehicle to dilate the time window available for steering to account for saturation of the steering actuator. Human-in-the-loop simulations with 17 volunteers and with 4 different control modes are carried out to assess performance enhancement. The framework simulates the vehicle teleoperation environment considering real-network-like variable delays and provides delayed visual feedback to the human operator accordingly.

Scenario 1 simulates a classical path-tracking use case. It consists of different regions with progressively increasing difficulty, e.g., wet terrain (region D) and strong cross wind (region E). Performance indices considered are related to cross-track error and steer effort. Laps traversed with SRPT mode shows evident performance enhancement compared to laps with other modes. This indicates better controllability and human–operator confidence.

Scenario 2 simulates bypassing a sudden obstacle and returning back to a straight path. Performance indices considered are related to net lateral movement and min safe distance observed from the obstacle. Evidently, in the case of sudden obstacles, without reducing speed, it is not possible to complete the track safely. Laps traversed with SRPT mode shows the maximum achieved safety distance and lesser net lateral movement compared to runs with other modes. Reduction in net lateral movement indicates better stability and improved operation safety.

Future work—Moving from virtual to real world. In this work, vehicle states fed to the NMPC block are considered known. First, in real-world experiments, a state estimator would be required to estimate vehicle states that are given in Equation (Equation 7). Its estimation accuracy is important for the success of the SRPT approach, e.g., over-estimation of lateral tire forces may lead the NMPC block to choose a conservative steer. Nevertheless, the IMU, steer encoders and speed encoders-based vehicle state estimator shows adequate accuracy for normal street-case maneuvers [33]. Second, the vehicle must be equipped with an emergency (stop) maneuver in case of failure of the sensor, actuator, or network. Third, the robustness of the SRPT approach must be assessed with variation of the model parameters, such as tire characteristics (due to wear), vehicle mass (and its distribution), road bank angle, etc. Indices used in the result section are applicable then also to assess performance improvement.

## Figures and Tables

**Figure 1 sensors-22-09119-f001:**
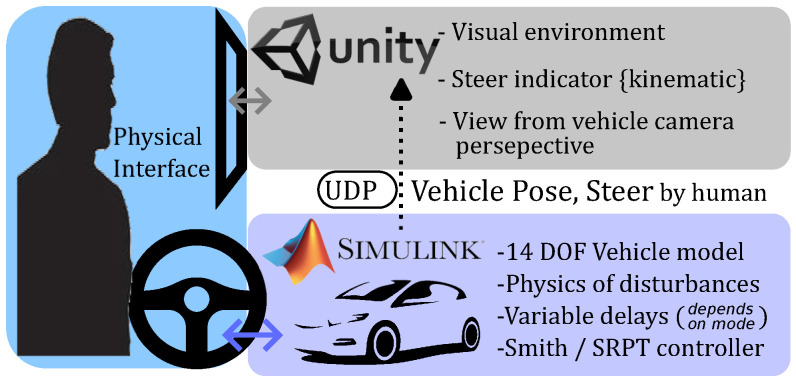
Integration of Simulink and Unity for human-in-loop vehicle teleoperation experiments.

**Figure 2 sensors-22-09119-f002:**
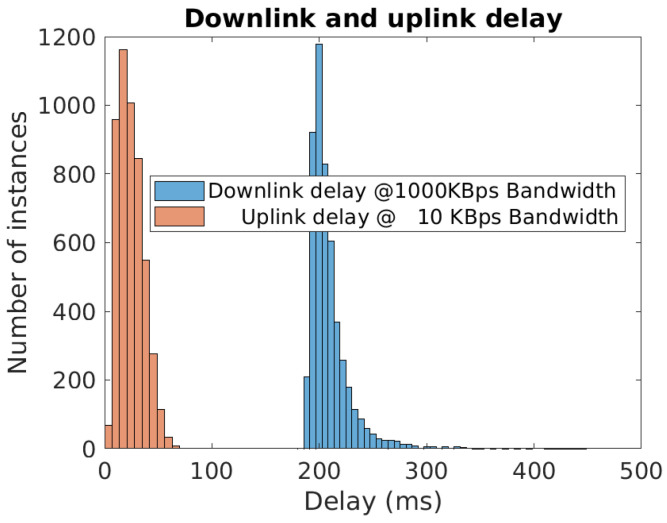
Delays observed in data transmission over 4G. Reprinted with permission from Ref. [23].

**Figure 3 sensors-22-09119-f003:**
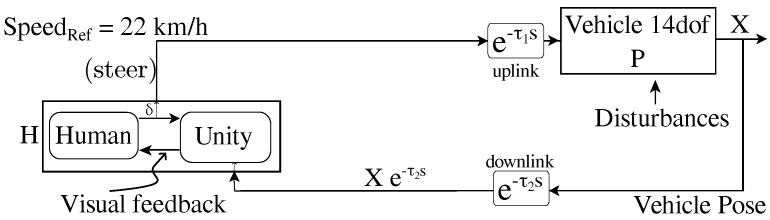
Schematic of typical vehicle teleoperation data flow.

**Figure 4 sensors-22-09119-f004:**
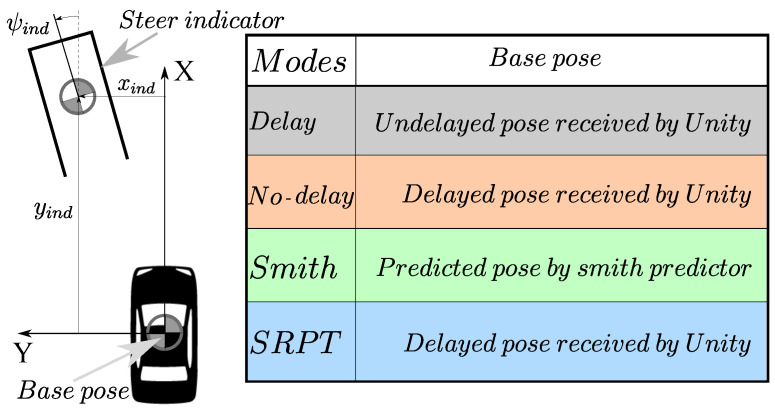
Steer indicator in respective modes.

**Figure 5 sensors-22-09119-f005:**
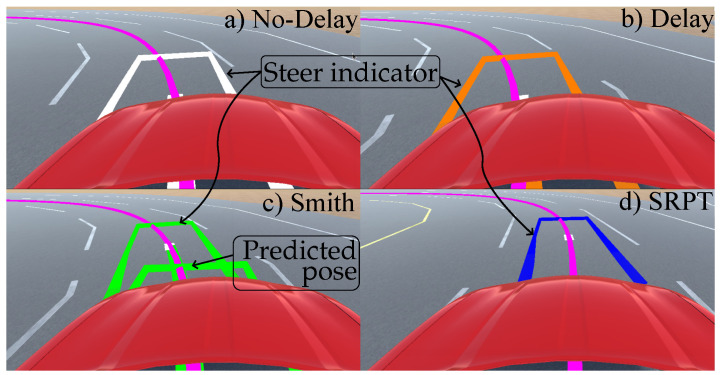
Vehicle perspective in (**a**) no-delay mode, (**b**) delay mode, (**c**) Smith predictor mode, and (**d**) SRPT mode.

**Figure 6 sensors-22-09119-f006:**
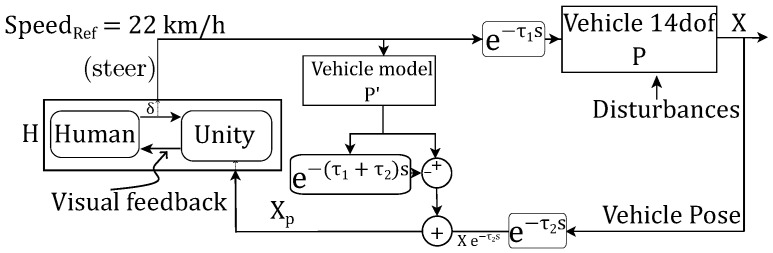
Smith-predictor scheme.

**Figure 7 sensors-22-09119-f007:**
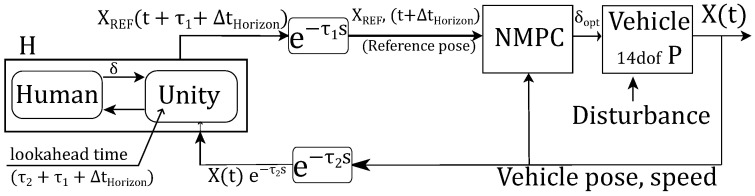
Successive reference-pose-tracking (SRPT) scheme.

**Figure 8 sensors-22-09119-f008:**
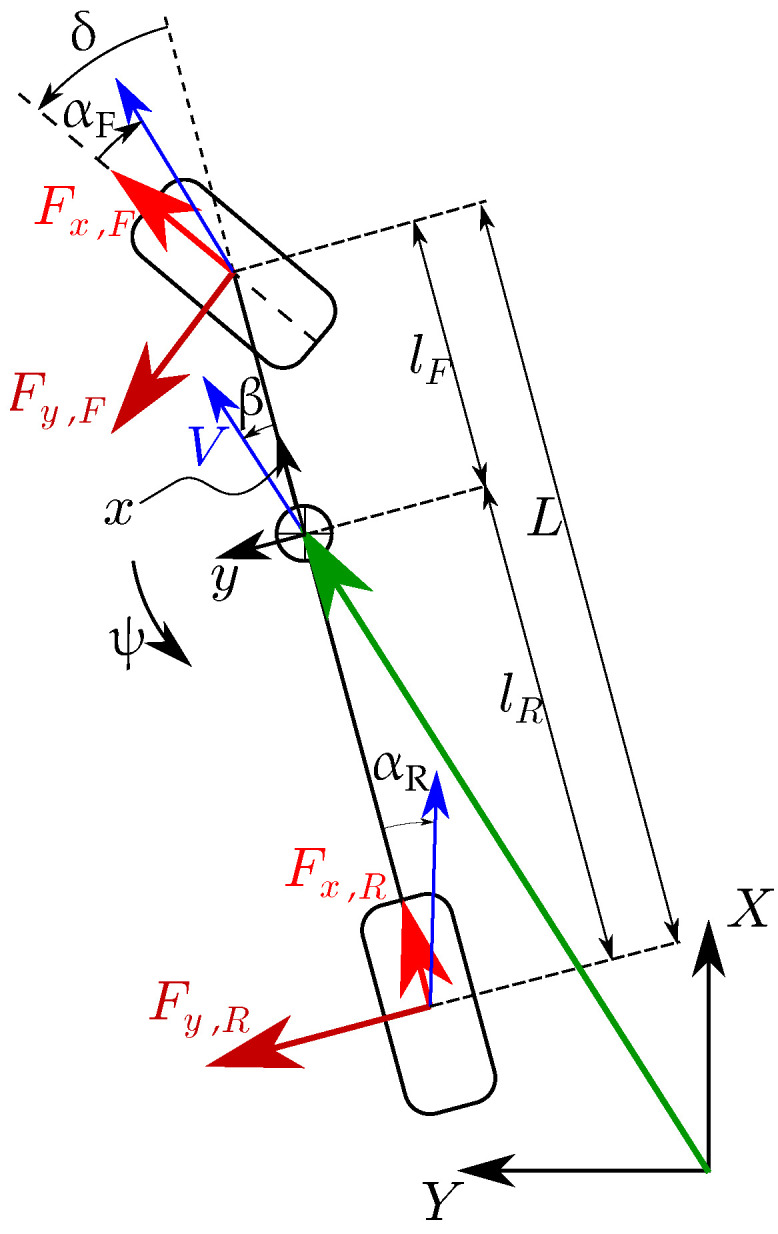
Single-track vehicle model. Reprinted with permission from Ref. [23].

**Figure 9 sensors-22-09119-f009:**
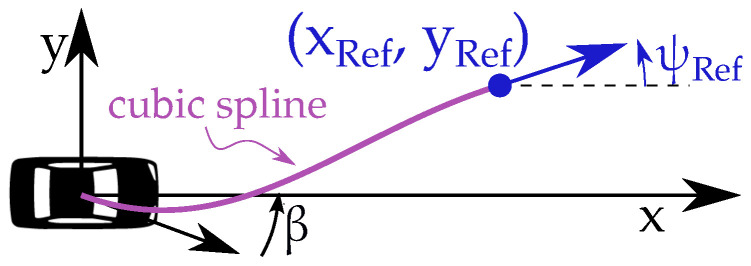
Cubic spline generation in vehicle reference frame.

**Figure 10 sensors-22-09119-f010:**
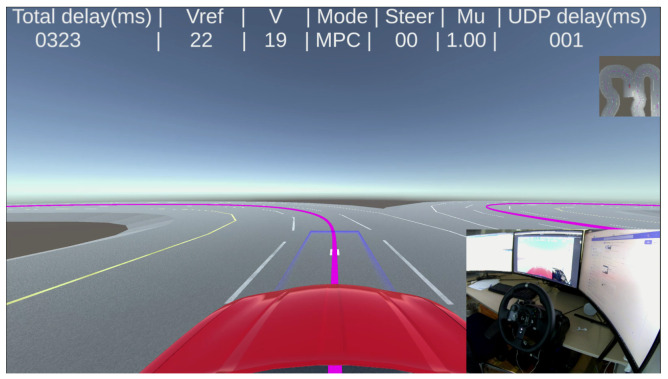
A snapshot of simulation in SRPT mode. Using steering joystick, human operator tries to keep the blue steer indicator on the purple trajectory. The NMPC controller inside the vehicle tries to follow the received reference-poses.

**Figure 11 sensors-22-09119-f011:**
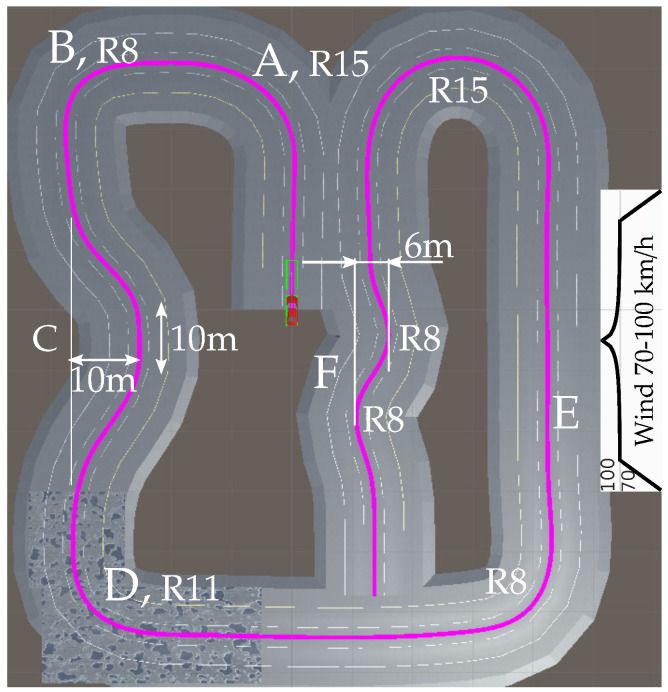
Scenario 1: Track with difficult maneuvers.

**Figure 12 sensors-22-09119-f012:**
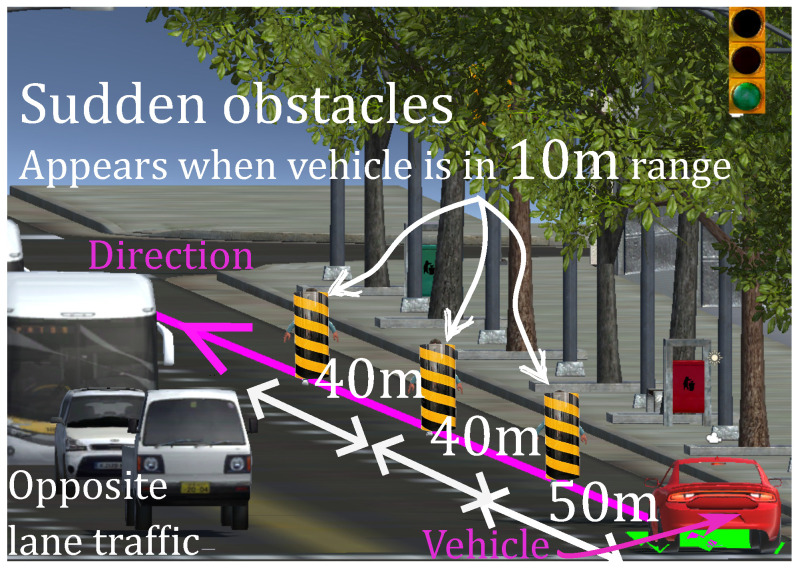
Scenario 2: Sudden obstacle avoidance in urban environment.

**Figure 13 sensors-22-09119-f013:**
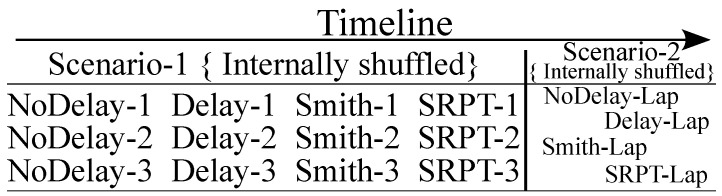
Randomization of test runs for each volunteer.

**Figure 14 sensors-22-09119-f014:**
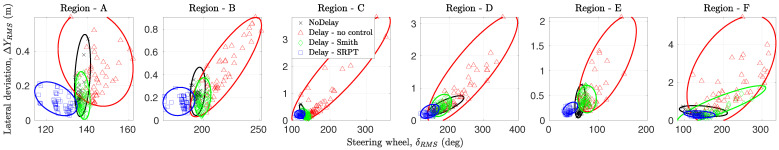
Scenario 1: Lateral deviation from the center line vs. steer by human at control station.

**Figure 15 sensors-22-09119-f015:**
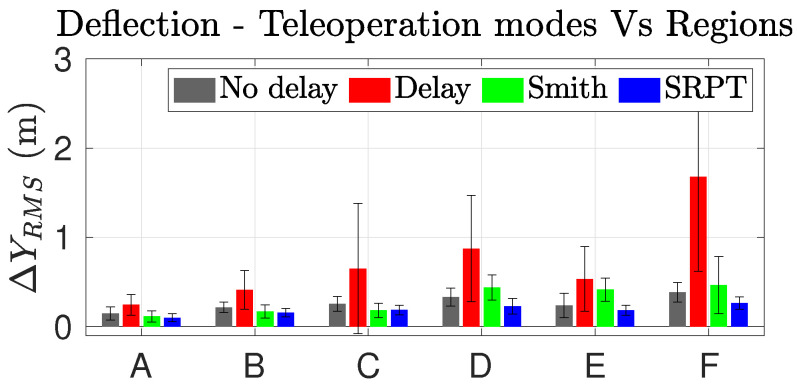
Lateral deviation in regions vs. modes.

**Figure 16 sensors-22-09119-f016:**
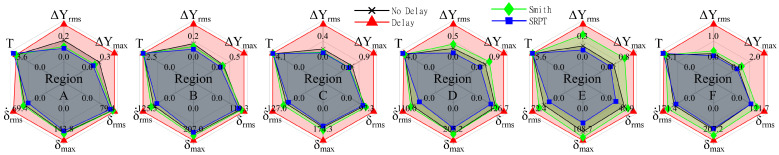
Scenario 1—Summary of performance indices.

**Figure 17 sensors-22-09119-f017:**
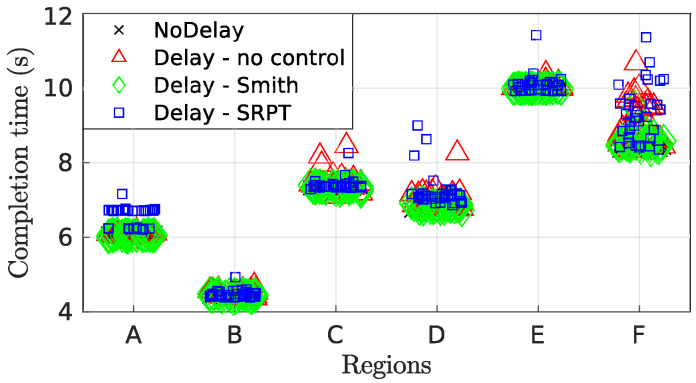
Scenario 1—Completion time for track regions.

**Figure 18 sensors-22-09119-f018:**
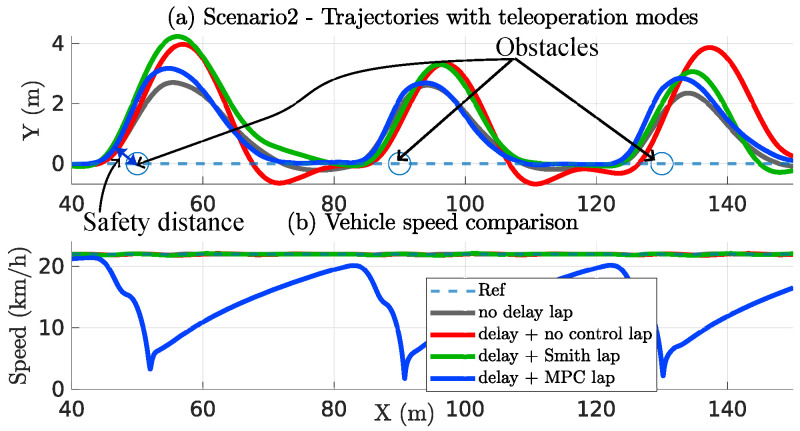
Scenario 2—Trajectories and vehicle speed profile.

**Figure 19 sensors-22-09119-f019:**
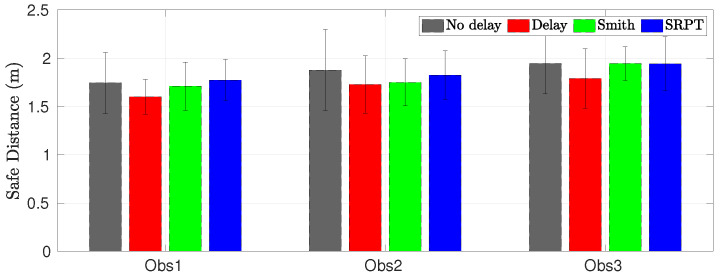
Scenario 2—Mean safe distance for all drivers.

**Figure 20 sensors-22-09119-f020:**
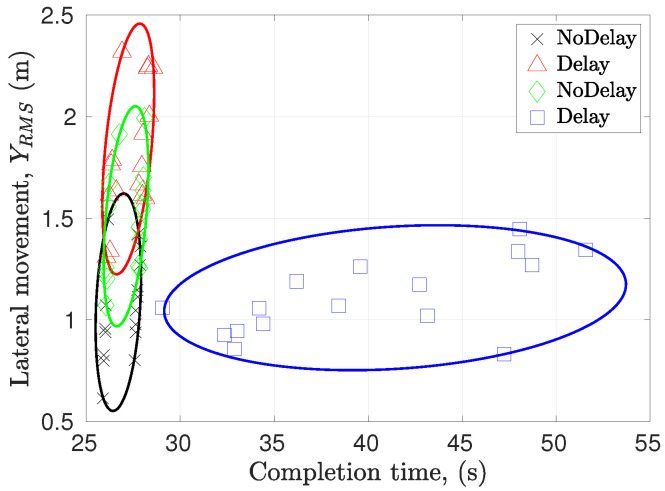
Scenario 2—Performance assessment summary. At the cost of higher completion time, SRPT mode resulted similar lateral movement as observed in No-delay mode. Lesser lateral movement indicates better controllability.

**Table 1 sensors-22-09119-t001:** Vehicle parameters for the single-track model.

Parameter	Value
*m*	1681kg
$I_{Z}$	2600 kg s^2^
$\left[m_{F};\;m_{R}\right]$	[871.6; 809.4] kg
$\left[l_{F};\;l_{R}\right]$	[1.3; 1.4] m
$\left[B_{x,F};\;B_{x,R}\right]$	[9.94; 10.6]
$\left[C_{x,F};\;C_{x,R}\right]$	[1.46; 1.46]
$\left[D_{x,F};\;D_{x,R}\right]$	[9643.4; 9019.0] N
$\left[B_{y,F};\;B_{y,R}\right]$	[9.8; 10.4]
$\left[C_{y,F};\;C_{y,R}\right]$	[1.29; 1.29]
$\left[D_{y,F};\;D_{y,R}\right]$	[8361.2; 7827.2] N
λ (Relaxation length)	0.3m
γ (Braking bias)	0.6
$C_{Aero}$ (Aerodynamic drag)	0.3675 N/(m^2^/s^2^)
$f_{V}$ (Rolling resistance coeff)	0.01

**Table 2 sensors-22-09119-t002:** NMPC cost penalties.

R	Q	P
diag([1,0.1])	0.1	diag([50,3])

**Table 3 sensors-22-09119-t003:** Performance improvement in $\Delta Y_{rms}$ and $\delta_{rms}$. SRPT mode performance is highlighted with blue background.

	$\Delta Y_{\mathrm{rms}}⇓$			$\delta_{\mathrm{rms}}⇓$		
**Region**	**No-** **Delay** **Mode**	**Smith** **Mode**	**SRPT** **Mode**	**No-** **Delay** **Mode**	**Smith** **Mode**	**SRPT** **Mode**
A	40%	54%	59%	6%	5%	12%
B	47%	59%	62%	10%	8%	16%
C	61%	71%	72%	26%	22%	31%
D	62%	50%	74%	25%	21%	37%
E	56%	22%	66%	27%	14%	52%
F	77%	72%	84%	38%	22%	39%

## Data Availability

Not applicable.

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
