# Peer review of "Vehicle Teleoperation: Human in the Loop Performance Comparison of Smith Predictor with Novel Successive Reference-Pose Tracking Approach"

_sensors, 2022, doi:10.3390/s22239119_

Round 1

Reviewer 1 Report

This paper discusses how to generate the reference trajectory for a teleoperated virtual vehicle with a teleoperated wheel steerer and compares the performance of the looped human trajectory tracking of the Smith predictor and the SRPT approach. Experiments with different types of drivers are performed on an advanced vehicle model in Simulink.

The paper is well written and the methodology is well presented, with an adequate introduction to the problem and coherently analyzed results. 

In my opinion it would be interesting to complete the study with an analysis of what to expect when moving from a virtual to a real vehicle and how the analysis means are expected to be altered in real operating situations.

Author Response

Thank you for reviewing our manuscript and for your valuable feedback.

As per your suggestion, the Introduction and conclusion sections are modified.

Introduction section:

The Outline of the paper is appended as shown below.

Conclusion section:

Aspects to be considered for real-world replication are added as shown below.

Reviewer 2 Report

none

Author Response

Thank you for reviewing our manuscript and for your valuable evaluation.

Reviewer 3 Report

The manuscript discusses an important topic of simulation of vehicle behavior in the virtual world and bringing human back into the loop to improve the quality of simulations. Such simulations reduce the number of real experiments, save resources and produce data for analysis. 

The manuscript is well written, it has a logical structure, and each section is important to introduce the research work. The introduction section provides a proper analysis of the field and highlights research goals. It follows with an analysis of related works that provide a literature review on the topic. The methodology section is well-written. Results are clearly presented and discussed in the conclusion section. 

Overall formatting and writing style are good. I suggest accepting current contribution to be a part of MDPI Sensors Special Issue Mechatronics Technology and Its Application in Intelligent Transportation.

Author Response

Thank you for reviewing our manuscript and for recommending acceptance of the manuscript for the journal.